# Spatial-Temporal Variation of N, P, and K Stoichiometry in Cropland of Hainan Island

Chunhua Ji [1,2], Hailin Liu [2], Zhengzao Cha [2], Qinghuo Lin [2] and Gu Feng [1,*]

1   College of Resources and Environmental Sciences, China Agricultural University, Beijing 100193, China; jchlover@163.com
2   China Rubber Research Institute, Chinese Academy of Tropical Agricultural Sciences, Haikou 571101, China; hlliuf@163.com (H.L.); chazhengzao@163.com (Z.C.); qinghuol@163.com (Q.L.)
*   Correspondence: fenggu@cau.edu.cn

**Abstract:** Elemental stoichiometry reflects the interaction between plants, soil, and microorganisms, and links biogeochemical patterns with physiological limitations. The stoichiometry of elements in farmland soil is an important part of the function of the agroecosystem. Soil nitrogen (N), phosphorus (P), and potassium (K) are the main macronutrients in terrestrial ecosystems, which are closely related to biogeochemical cycles. Studying the temporal and spatial variability of soil nutrients in tropical farmland is of great significance for exploring the variation of soil nutrients and promoting the sustainable development of tropical agriculture. In this study, soil samples in the farmland of Hainan Island were collected at three different stages for exploring temporal and spatial variations of N, P, and K stoichiometry. Results showed that soil concentrations of available N, P, and K changed markedly with the temporal and spatial variability. The highest available N, P, and K concentrations appeared at the stage of 2016–2020 with values of 110.40 mg/kg, 51.91 mg/kg, and 82.76 mg/kg, respectively, while their lowest values were observed in 2010–2015 with 66.34 mg/kg, 11.27 mg/kg, and 45.77 mg/kg, respectively. The available nitrogen content in the three time periods first increased and then decreased with the increase of available potassium content, an opposite trend was observed between available nitrogen and phosphorus. The content of N increased in Haikou, Lingao, Ding'an, and P increased in Wengchang, and Lingshui and K increased in Danzhou and Wanning as time increased.

**Keywords:** Hainan Island; soli nutrient; spatial-temporal variability; stoichiometry

## 1. Introduction

Stoichiometry provides a framework approach to understanding the function of ecosystems and the balance of elements [1]. The development and suitability of plants largely depend on the availability of mineral elements in the soil [2]. Nitrogen (N), phosphorus (P), and potassium (K) as important mineral elements are one of the core focuses of ecosystem ecology [3–5].

They also directly affect soil microbial dynamics, soil organic carbon, and the long-term accumulation of soil nutrients [6–8]. After carbon, nitrogen is the element most needed by crops because it is a component of protein, chlorophyll, and many metabolic reactions [9]. At the same time, phosphorus and potassium are very important for the growth and physiological response of plants in low-fertility soils [10]. There are many studies on the stoichiometry of nitrogen and phosphorus in terrestrial ecosystems, but some studies show that potassium is more correlated with the stoichiometry of different plant ecotypes than nitrogen or phosphorus [11,12]. However, potassium, as the third important element for plant production after nitrogen and phosphorus, has many important roles in terrestrial biogeochemical cycles that have been neglected [13]. Moreover, the current reports on potassium are relatively lacking, and most of them focus on the research of plant

leaves [14,15]. Therefore, this study will study the stoichiometry of potassium in addition to nitrogen and phosphorus, in order to provide more comprehensive information on the stoichiometry of macronutrient elements.

In agricultural systems, the natural biogeochemical cycle can be supplemented by adding ready-made inorganic fertilizers [8]. Human activities may cause imbalances in nutrient interaction and even change the nutrient cycle [16,17]. Additionally, the quantity and supply capacity of farmland soil nutrients are of great significance to human nutrition acquisition and the global biogeochemical cycle [18]. Therefore, it is very meaningful to study the characteristics of stoichiometry in farmland soil. However, at present, most studies on stoichiometry focus on plant leaves, mainly involving different succession stages, different community types, and different intervention methods, among which the most extensive studies have focused on the stoichiometric characteristics of N, P, and N/P in plant leaves [14,15]. The study of soil stoichiometry is of great significance for analyzing the stability of the ecosystem, the limitation of soil nutrients, and the strategy of plant adaptation to the environment [19]. The stoichiometry of soil N, P, and K can not only reflect the fertility status of the soil, but its ratio is an important indicator of the soil organic matter composition, soil quality status, and nutrient supply capacity, as well as the index of soil N and P mineralization and fixation [20]. Through the change law of the ratio of soil N, P, and K can not only effectively provide theoretical guidance for soil nutrient management but also help determine the response of element changes in the ecosystem to global environmental changes and the N cycle process [21]. Considering the importance of soil stoichiometry research, this research focuses on the analysis of macronutrient elements such as available nitrogen, phosphorus, and potassium in farmland soils in Hainan Island.

Although some studies have gradually applied ecological stoichiometry to crop soils in recent years, these studies have mainly focused on the effects of different fertilization methods [22,23], different environmental stresses [24], and different growth stages [25] on the ecological stoichiometry of crops. There is a lack of research on farmland soil stoichiometry that includes both temporal and spatial variation. The spatial variability of soil properties is not only an important content of soil science research but also an important aspect of sustainable land management [26]. Moreover, a detailed understanding of the spatial variability of soil is the basis for the realization of refined agricultural management [27]. The precision agriculture advocated by people now uses information technology to achieve efficient management of agriculture based on the temporal and spatial differences of farmland including soil nutrients [28]. Therefore, precise temporal and spatial differences in farmland are of great significance to the realization of precision agriculture. Moreover, the spatial distribution of soil nutrients in cultivated land is the basis for improving the productivity of cropland [29]. In addition, the high spatial heterogeneity of soil nutrients, coupled with the limitation of observation, causes great uncertainty in the study of spatial variation of soil nutrients stoichiometry [30]. Therefore, the study of soil spatial variability will always be the focus of the study of soil properties. It is very important to study the temporal variability of elements in the soil because temporal analysis helps to fully explain the functions of the soil [31,32]. Therefore, the study of temporal and spatial variation of farmland soil nutrient stoichiometry is helpful to understand the controlling factors of soil nutrient stoichiometry on a larger scale. In the study of soil spatial-temporal variability, it has been shown that the temporal and spatial distribution of soil moisture is affected by many factors such as precipitation, vegetation coverage, and soil properties [33,34]. However, the research on the temporal and spatial variability of soil stoichiometry is relatively lacking. Therefore, this study aims to deepen the understanding of the temporal and spatial variation of farmland soil stoichiometry through the analysis of N, P, and K stoichiometry in farmland soil of Hainan Island, and contribute to a deeper analysis under a larger ecological background. The study was of great significance for exploring the variation of soil nutrients and promoting the sustainable development of tropical agriculture. Our hypotheses are: (1) there are differences in the element content

of N, P, and K under different time and space conditions; (2) the relationship and spatial variation law of available N, P, and K are similar in different time periods.

## 2. Method

### 2.1. Study Site

Hainan Island is located at the southernmost tip of China (108°37′–111°03′ E, 18°10′–20°10′ N) with a land area of 33,920 square kilometers. The region has a tropical monsoon climate, with an average annual rainfall of 1600–2500 mm and an annual average temperature of 23–25 °C [35]. The main soil types of Hainan Island include ferrisols, ferralsols, alfisols, cambosols, andosols, anthrosols, halosols, and entisols [36].

Because the terrain of Hainan Island is medium-high and low-period, regardless of soil-forming conditions or soil distribution, it gradually changes to the surroundings with the central and southern mountains as the center. The soil distribution is extremely affected by topography. The soil of the whole island is distributed into several concentric circles, which are distributed in sequence around the central mountains: the outermost circle is the coastal sandy soil distributed around the modern coastal terraces of the whole island, and its altitude is only within 1 m; the outer ring is a zonal soil-brick red soil, which is mainly distributed on terraces, terraces, and hills below 350 m (or 400 m) above sea level. The soil zone in the northern part of the island is wider, and the soil zone in the southern part of the island is narrower. The mountains in the south-central part of the island are yellow soil, with an altitude of more than 400 m [37].

### 2.2. Sample Collection and Determination

The geographical locations of the sample points were located by GPS. The longitude and latitude were recorded in detail. From 2006 to 2020, soil samples were collected annually, and a total of 5877 soil samples were collected. The distribution of sampling sites took into account the land use of Hainan Island. At each sampling site, the surface soil of 0–20 cm was selected to collect five separate farmland soil samples, and then the five separate soil samples were fully mixed and 1 kg soil samples were selected according to the quartering method for testing.

The air-dried soil samples were passed through 2 mm sieves and then the sieved soil samples were mixed uniformly and put into a zip lock bag. The soil available nitrogen was determined by the alkaline hydrolysis distillation method. Soil available P was measured by the colorimetric method after hydrochloric acid-ammonium fluoride extraction. Available K was determined using flame photometry after ammonium acetate extraction [38,39].

### 2.3. Data Analysis

We used the average values of elements in the three time periods from 2006 to 2010, 2011 to 2015, and 2016 to 2020 to represent the element concentrations in these time periods. We also made a significant comparison of different element concentrations in different time periods. At the level of $p < 0.05$, the concentration of N, P, and K in three time periods was analyzed by one-way analysis of variance (ANOVA), and Tukey's multiple comparison method was used to test the significance of the difference. The density histogram was used to study the frequency and the range of data. To improve normality of distributions, the data of the content N, P, and K and the ratio of the N, P, and K were log10-transformed (Figures 3 and 4) for the permutation test, but shown in untransformed values for easier understanding in our article. The element concentration was converted to Ln, and then the quadratic regression analysis was performed. The density histogram was used to study the influence of different time periods on the content of farmland soil N, P, and K. All the figures were generated using EXCEL 2016 or SPSS 25.0. The spatial distribution map of available nitrogen, phosphorus, and potassium is made using ArcMap 10.2.

## 3. Result

### 3.1. Variation of Stoichiometry of N, P, and K with Time

Here, we measured the stoichiometric ratio of N, P, and K (Figure 1) for all years. The content of available nitrogen, phosphorus, and potassium all showed an increasing trend after 2015. No obvious trend is observed between N/P and N/K in each year, and further analysis and verification are needed. P/K also showed an increasing trend since 2015. We also measured available macronutrient (N, P, K) contents in the cropland of Hainan Island from 2006 to 2020 and classified them every five years (2006–2010, 2011–2015, 2016–2020).

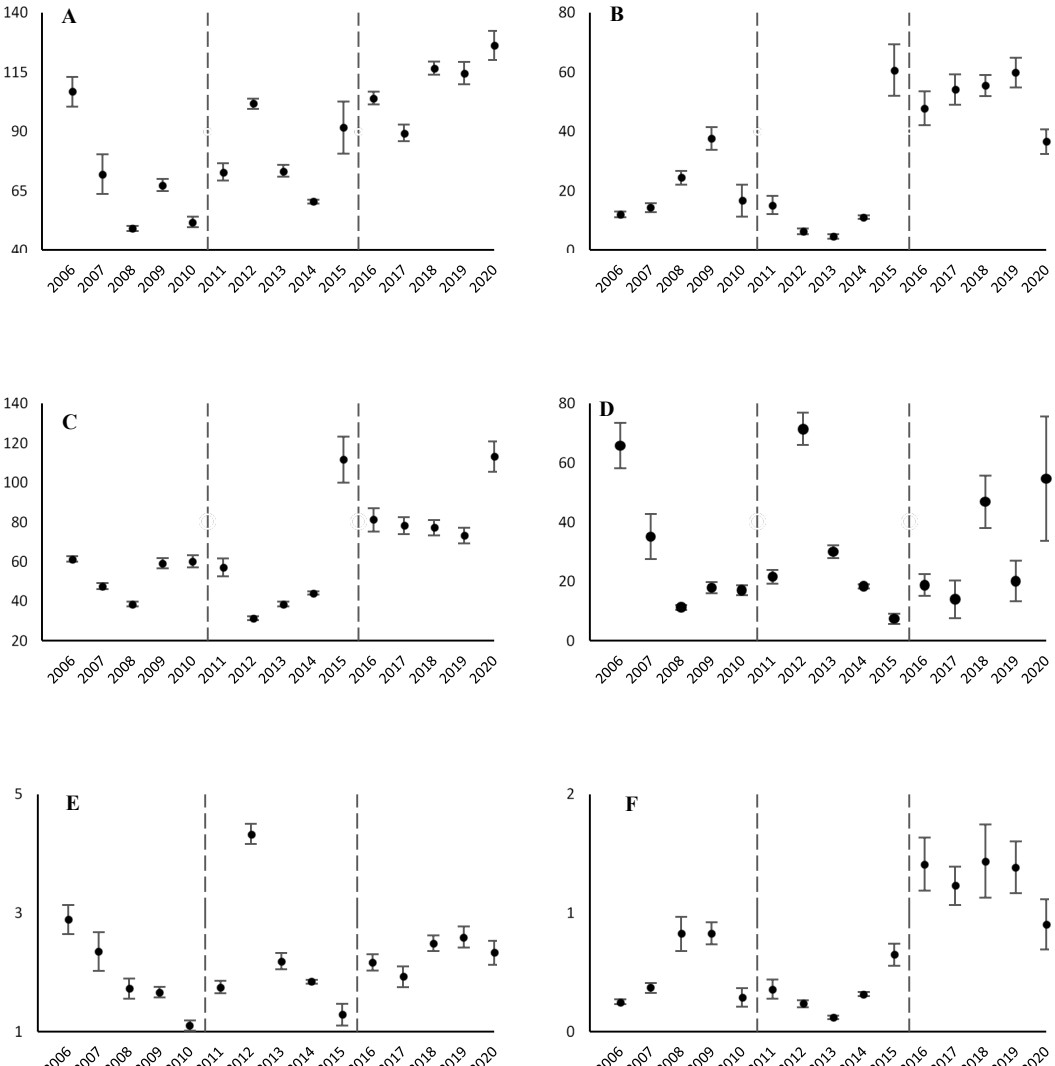

**Figure 1.** Annual N, P, and K stoichiometry and ratios. The solid black circles indicate the average value for that year and the lines indicate the value of SE. SE is derived based on the annual data volume of N, P, and K. The content of available nitrogen (**A**) at all years. The content of available phosphorus (**B**) at all years. The content of available potassium (**C**) at all years. The stoichiometric ratio of N and P (**D**) at all years. The stoichiometric ratio of N and K (**E**) at all years. The stoichiometric ratio of P and K (**F**) at all years.

The variation of the stoichiometric properties of N, P, and K with time is shown in Figure 1. The available nitrogen was 75.93 mg/kg, 66.34 mg/kg, and 110.40 mg/kg at different stages of 2006–2010, 2011–2015, and 2016–2020, respectively (Figure 2A). The available phosphorus was 27.28 mg/kg, 11.27 mg/kg, and 51.91 mg/kg at different stages of 2006–2010, 2011–2015, and 2016–2020, respectively (Figure 2B). The available potas-

sium was 64.88 mg/kg, 45.77 mg/kg, and 82.76 mg/kg at different stages of 2006–2010, 2011–2015, and 2016–2020, respectively (Figure 2C). Additionally, the concentrations of available nitrogen, phosphorus, and potassium at different stages presented a similar trend with the significant higher nutrient element concentration in 2016–2020 than that in the other two stages, and the significantly lowest element concentration occurred in 2011–2015 (Figure 2A–C). However, the ratio between N, P, and K did not work that way. No significant differences were found in N/P across the three time stages (Figure 2D). N/K showed a different pattern, which was significantly the largest in 2016–2020 (2.33) (Figure 2E). The average value of P/K in 2006–2010, 2011–2015, and 2016–2020 were 0.6, 0.3, and 1.3, respectively. Moreover, the P/K in 2016–2020 was significantly highest, and the P/K in 2010–2015 was significantly lowest.

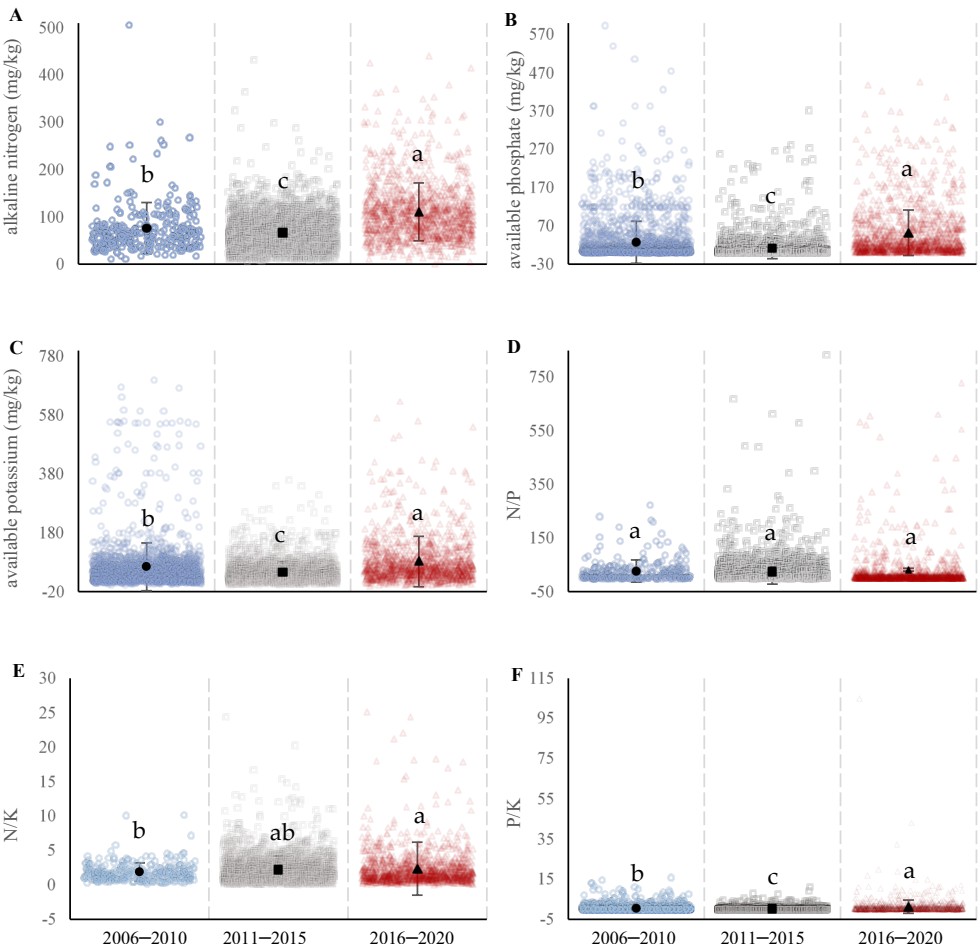

**Figure 2.** The stoichiometric ratio of N, P, and K at different stages. The blue open cycles represent the concentration of elements in 2006–2010, the gray open squares represent the concentration of the in 2011–2015, the red open triangles represent the N, P, and K concentration in 2016–2020. The black dots represent the average value of N, P, and K concentrations in this time period. Line bars show the value of SE. Letters above the line bars show the results of Tukey tests. Slopes with the same letters are not significantly different, while those with different letters are significantly different. The content of available nitrogen (**A**) at different stages. The content of available phosphorus (**B**) at different stages. The content of available potassium (**C**) at different stages. The stoichiometric ratio of N and P (**D**) at different stages. The stoichiometric ratio of N and K (**E**) at different stages. The stoichiometric ratio of P and K (**F**) at different stages.

Available nitrogen, phosphorus, and potassium all vary greatly in the three time spans. Available nitrogen ranged from 10.9 mg/kg to 505.7 mg/kg, from 6.3 mg/kg to 432.2 mg/kg, and from 1.1 mg/kg to 440.7 mg/kg for 2006–2010, 2011–2015, and 2016–2020, respectively (Figure 3A–C); available phosphorus from 0.3 mg/kg to 593.9 mg/kg, from 0.2 mg/kg to 372 mg/kg, and from 0.1 mg/kg to 676 mg/kg, respectively (Figure 3D–F); available potassium from 3 mg/kg to 700 mg/kg, from 3 mg/kg to 360 mg/kg, and from 1 mg/kg to 627 mg/kg, respectively (Figure 3G–I); the ratio of nitrogen to phosphorus from 0.1 mg/kg to 274.09 mg/kg, from 0.066 mg/kg to 835 mg/kg, and from 0.0073 mg/kg to 3706 mg/kg, respectively (Figure 4A–C); the ratio of nitrogen to potassium from 0.15 mg/kg to 10.15 mg/kg, from 0.039 mg/kg to 24.33 mg/kg, and from 0.02 mg/kg to 25.1 mg/kg, respectively (Figure 4D–F); the ratio of phosphorus to potassium from 0.002 mg/kg to 15.85 mg/kg, from 0.006 mg/kg to 11.04 mg/kg, and from 0.0006 mg/kg to 104.9 mg/kg, respectively (Figure 4G–I).

### 3.2. Spatial Variation of Available N, P, and K Stoichiometry

The available nitrogen content of Hainan Island was less than 30 mg/kg in the part of Lingao during the period of 2006–2010 (Figure 5A). During the period of 2006–2010, the available nitrogen content in some areas of Danzhou, Haikou, Chengmai, Ding'an, and Baoting ranged from 30 to 60 mg/kg. In 2006–2010, the available nitrogen content in parts of Changjiang, Ledong, and Wanning was 120–150 mg/kg. The areas with available nitrogen content greater than 150 mg/kg were mainly Baisha, Qiongzhong, Wuzhishan, and Baoting. In the remaining areas, the available nitrogen content was in the range of 60–90 mg/kg and 90–120 mg/kg, except for Dongfang City, where the available nitrogen content was 90–120 mg/kg. During the period of 2011–2015, the regions with available nitrogen content below 30 mg/kg included Dongfang, Ledong, and Danzhou, and the regions with an available nitrogen content of 30–60 mg/kg included Dongfang, Ledong, Danzhou, Tunchang, Baisha, Qiongzhong, Wuzhishan, Baoting, Lingshui, Qionghai, Chengmai, Wenchang, and Changjiang, and the regions with an available nitrogen content of 90–120 mg/kg included Lingao, Danzhou, Qiongzhong, Wanning, Qionghai, Tunchang, and Ding'an, Haikou, and the areas with an available nitrogen content of 120–150 mg/kg were mainly parts of Qiongzhong. The available nitrogen content in the remaining areas was 60–90 mg/kg. From 2016 to 2020, the area with 30–60 mg/kg, 60–90 mg/kg, and 90–120 mg/kg available nitrogen in farmland soil of Hainan Island were Danzhou, Baisha, and Changjiang, and the area with 90–120 mg/kg and 120–150 mg/kg available nitrogen was Tunchang and Baoting. The areas containing 60–90 mg/kg, 90–120 mg/kg, and 120–150 mg/kg were Wenchang, Dingan, Linggao, Chengmai, Qionghai, Wanning, Wuzhishan, Dongfang, and Ledong. The areas containing 90–120 mg/kg, 120–150 mg/kg, and >150 mg/kg were Haikou, Qionghai, Qiongzhong, and Sanya.

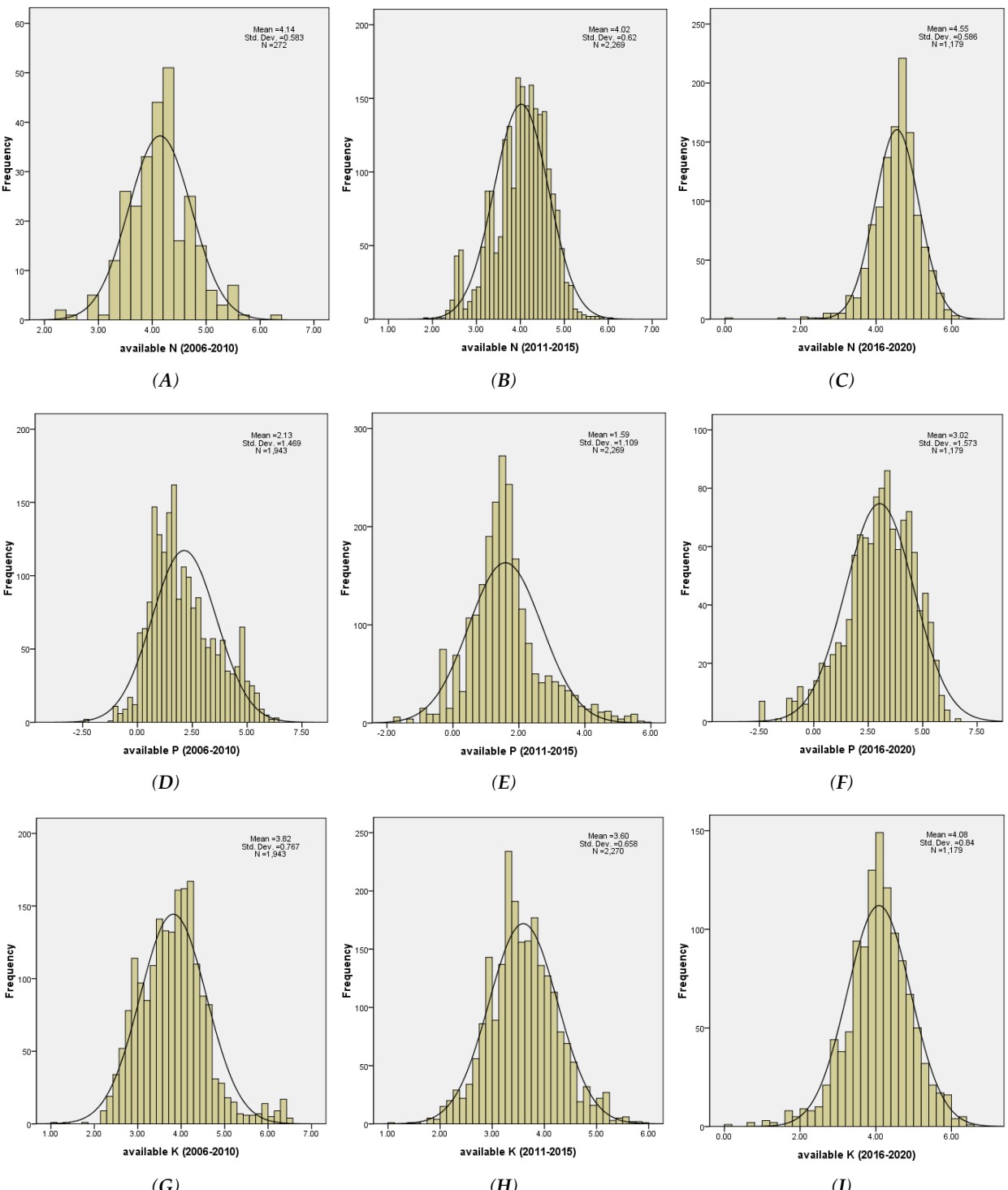

**Figure 3.** The density histogram of available nitrogen (**A**) for the 2006–2010 time period. The density histogram of available nitrogen (**B**) for the 2011–2015 time period. The density histogram of available nitrogen (**C**) for the 2016–2020 time period. The density histogram of available phosphorus (**D**) for the 2006–2010 time period. The density histogram of available phosphorus (**E**) for the 2011–2015 time period. The density histogram of available phosphorus (**F**) for the 2016–2020 time period. The density histogram of available potassium (**G**) for the 2006–2010 time period. The density histogram of available potassium (**H**) for the 2011–2015 time period. The density histogram of available potassium (**I**) for the 2016–2020 time period.

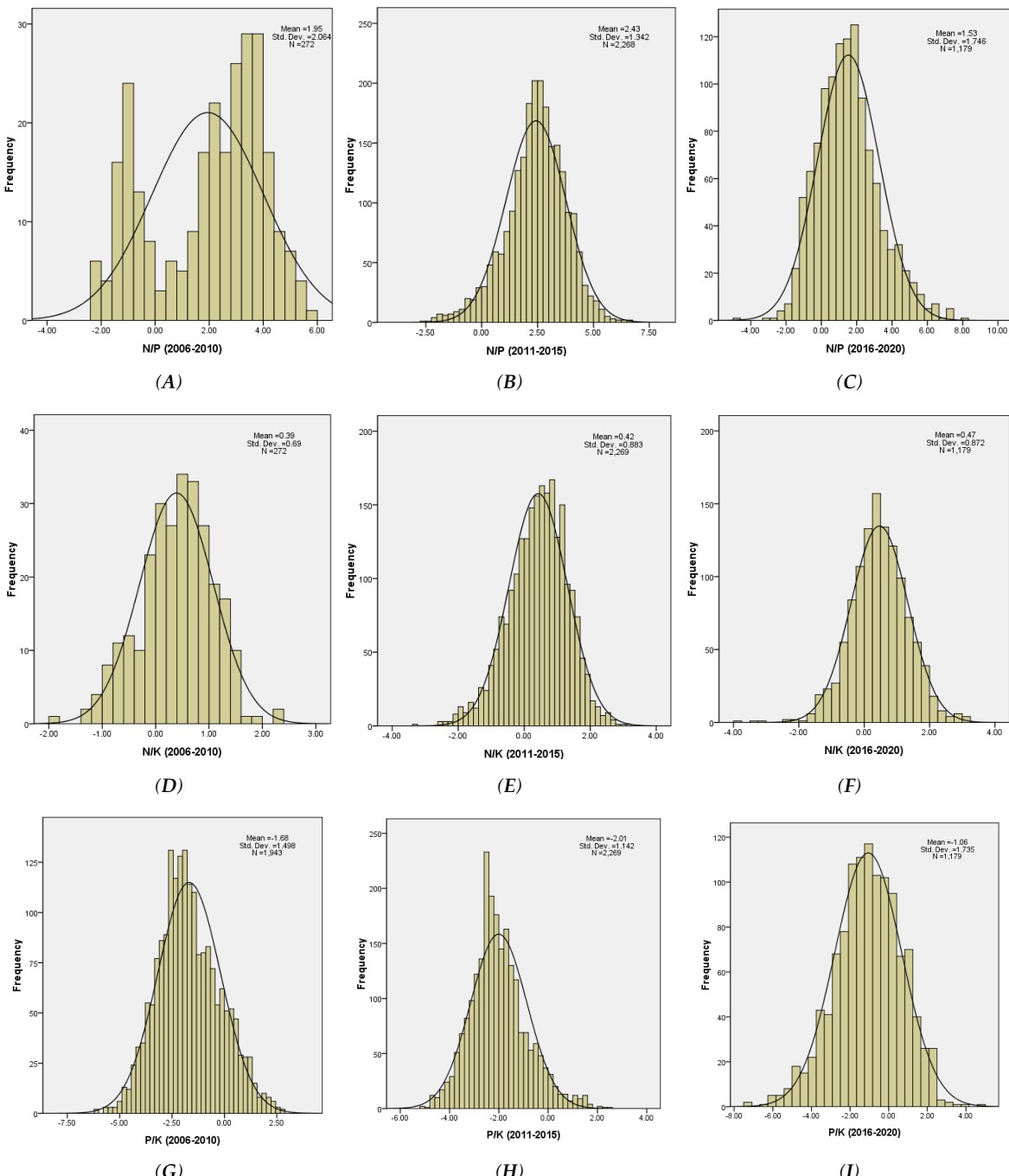

**Figure 4.** The density histogram of the ratio of N and P (**A**) for the 2006–2010 time period. The density histogram of the ratio of N and P (**B**) for the 2011–2015 time period. The density histogram of the ratio of N and P (**C**) for the 2016–2020 time period. The density histogram of the ratio of N and K (**D**) for the 2006–2010 time period. The density histogram of the ratio of N and K (**E**) for the 2011–2015 time period. The density histogram of the ratio of N and K (**F**) for the 2016–2020 time period. The density histogram of the ratio of P and K (**G**) for the 2006–2010 time period. The density histogram of the ratio of P and K (**H**) for the 2011–2015 time period. The density histogram of the ratio of P and K (**I**) for the 2016–2020 time period.

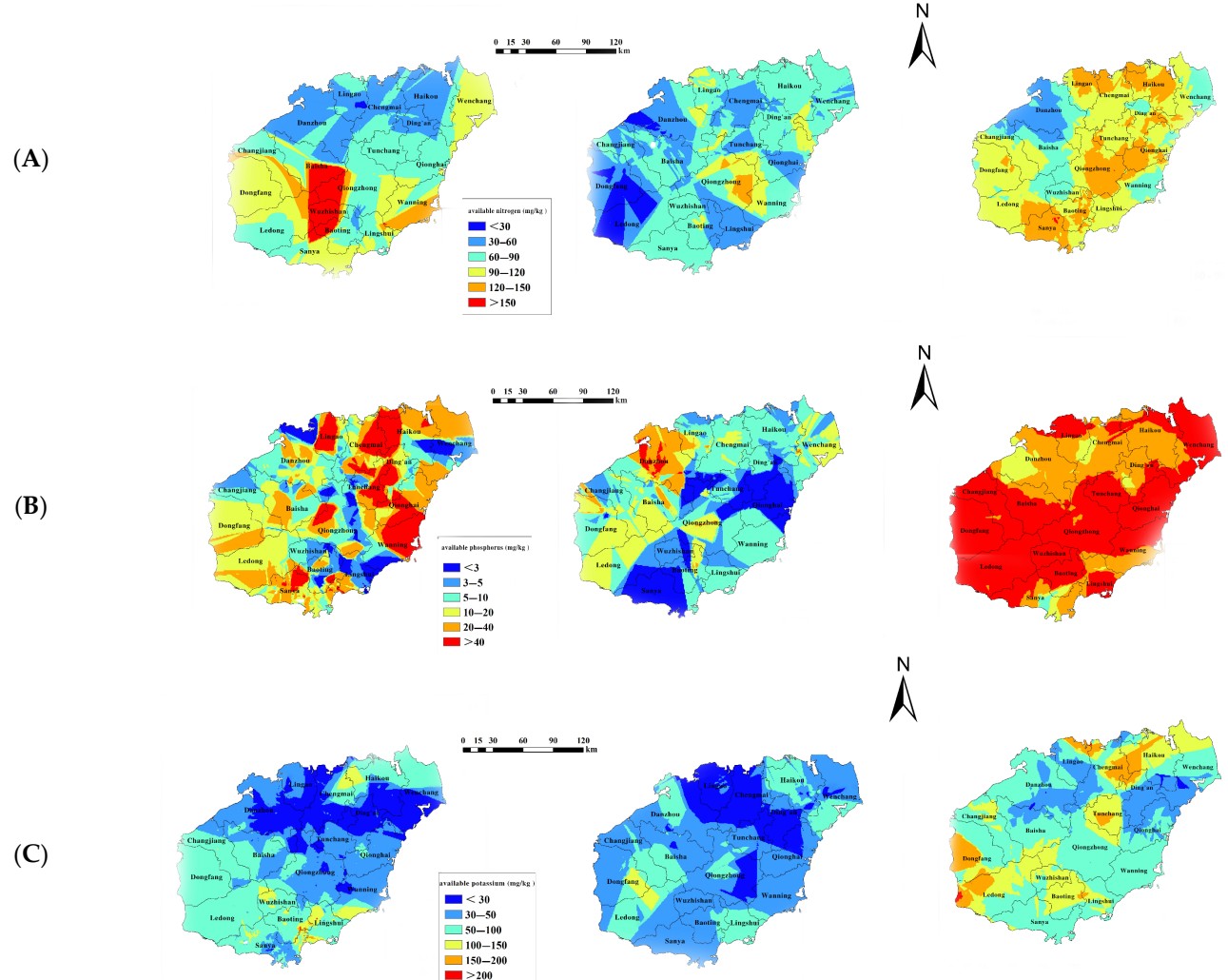

**Figure 5.** The spatial distribution of each indicator available nitrogen (**A**) at different stages. The spatial distribution of each indicator available phosphorus (**B**) at different stages. The spatial distribution of each indicator is available potassium (**C**) at different stages.

Figure 5B showed the spatial distribution characteristics of available phosphorus content in Hainan Province. The content of available phosphorus in the middle of Lingao, the west of Haikou, the east of Chengmai, parts of Sanya, the southwest of Qionghai, and the border area of Tunchang and Ding'an during 2006–2010 was more than 40 mg/kg. In 2006–2010, the content of available phosphorus was 20–40 mg/kg in the southern of Ledong and Dongfang, the western of Baisha, the central part of Qiongzhong, the northeastern of Qiongha, Sanya, Wenchang, and the northern of Haikou the Sanya, and in Dongfang, Ledong, northern Baisha, the central of Danzhou and Tunchang and Ding'an was 10–20 mg/kg, in northern Changjiang, northwest and east Danzhou, southern Haikou, and central Hainan was 5–10 mg/kg, while the content of available phosphorus in northern Danzhou, northwest Wenchang, and southern Lingshui was extremely low, less than 3 mg/kg. From 2011 to 2015, the content of available phosphorus in Danzhou was extremely high, both greater than 20 mg/kg, and even higher than 40 mg/kg in the central part of Danzhou. the content of available phosphorus was 10–20 mg/kg in Dongfang, Baisha, Qiongzhong, Baoting, Ledong, Chengmai, and part of Wenchang. The content of available P in Tunchang, Baisha, Qionghai, Ding'an, Baoting, and Sanya was less than 3 mg/kg, and in other areas of Hainan Island was 10–20 mg/kg. In 2016–2020, except for Chengmai,

Haikou, Ding'an, and Danzhou, the available phosphorus content in the soil of Hainan Province was greater than 40 mg/kg.

In 2006–2010, the content of available potassium in farmland soil of Hainan Island was higher in the southwest and lower in the east. Specifically, the soil available potassium content in southern Lingao, southern Danzhou, southwestern Chengmai, southern Haikou and Wenchang, and most areas of Ding'an were all less than 30 mg/kg. In western Danzhou, Baisha, Qiongzhong, Wanning, and Qionghai, soil available potassium content was 30–50 mg/kg. Soil available potassium content was 50–100 mg/kg in northern Wenchang, Haikou, Changjiang, Dongfang, Ledong, Sanya, and Lingshui, and 100–150 mg/kg in Haikou, Wuzhishan, Baoting, Sanya, and some parts of Lingshui. In 2010–2015, the content of available potassium was between 100 and 150 mg/kg in Ledong and parts of Dongfang, in Dongfang, Ledong, Lingshui, Qiongzhong, Danzhou, Haikou, and parts of Wenchang were 50–100 mg/kg, and less than 30 mg/kg in other regions. In 2016–2020, the content of available potassium in the soil of Haikou, Chengmai, Dongfang, and the northwest of Ledong was 150–200 mg/kg, and in eastern Ledong, Wuzhishan, Baoting, Lingshui, Tunchang was 100–150 mg/kg, and in Wenchang, Wanning, Qiongzhong, Baisha, Danzhou, and Sanya was 50–100 mg/kg (Figure 5C).

### 3.3. The Relationship of Available Nitrogen, Phosphorus, and Potassium on Time Scale

The available N, P, and K showed different relationships at different stages. The concentration of available nitrogen decreased first and then increased with the increase of available phosphorus concentration during 2006–2010, 2011–2015, and 2016 to 2020 (Figure 6A–C). During the period from 2006 to 2010, the content of available phosphorus increased significantly with the increase of available potassium content in the agricultural soil of Hainan Island (Figure 6D). In 2011–2015, the content of available P decreased first and then increased with the increase of available potassium content. When the content of available potassium was 2.3 mg/kg, the content of available P was 1.27 mg/kg at the lowest (Figure 6E). When the concentration of available potassium was less than 3.13 mg/kg, the concentration of available phosphorus from 2016 to 2020 decreased increased significantly with the increase of available potassium concentration; while when the available potassium concentration was greater than 3.13 mg/kg, the available phosphorus concentration increased significantly with the increase of available potassium concentration (Figure 6F). The available nitrogen concentration showed a trend of first increasing and then decreasing trend with the increase of the available potassium concentration in the three time periods of 2006–2010, 2011–2015, and 2016–2020 (Figure 6G–I).

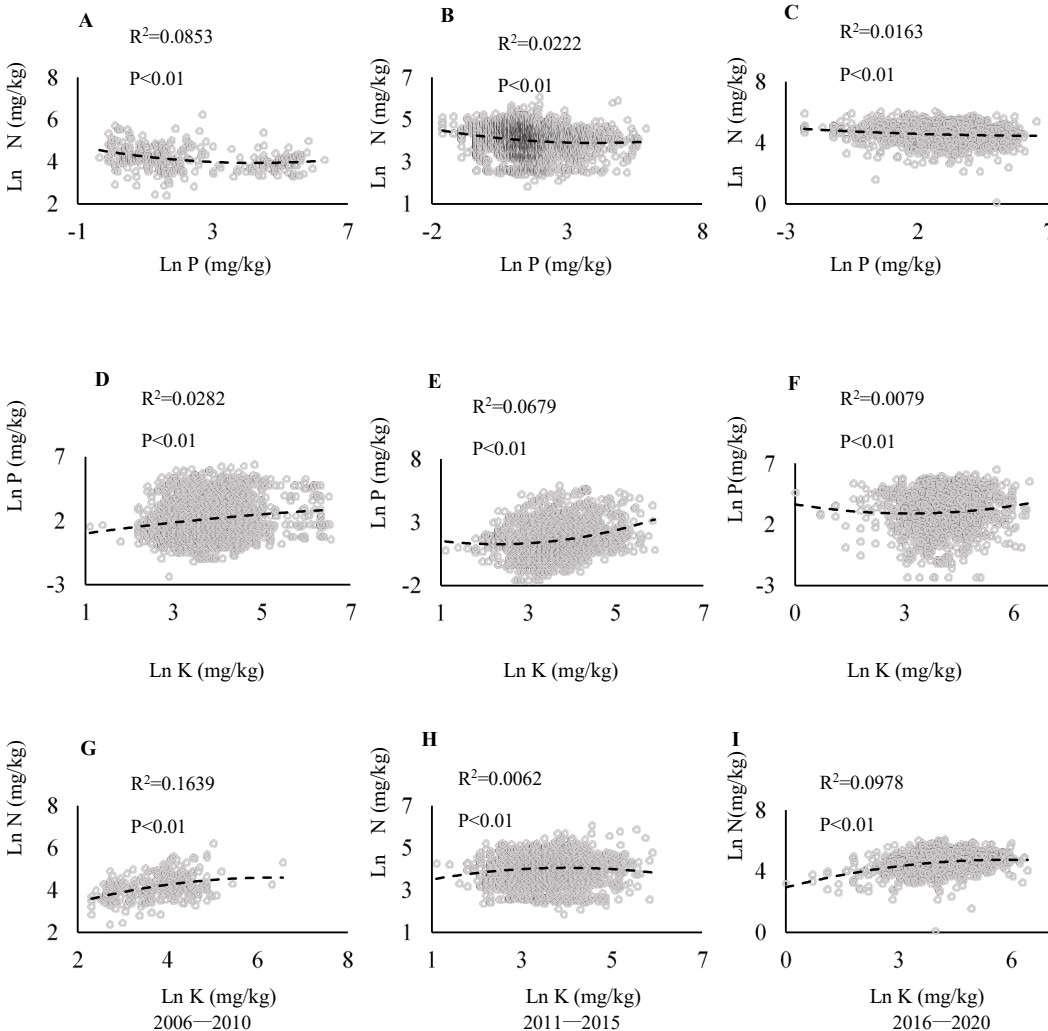

**Figure 6.** The relationship between available nitrogen and available phosphorus (**A**) for the 2006–2010 time period. The relationship between available nitrogen and available phosphorus (**B**) for the 2011–2015 time period. The relationship between available nitrogen and available phosphorus (**C**) for the 2016–2020 time period. The relationship between available phosphorus and available potassium (**D**) for the 2006–2010 time period. The relationship between available phosphorus and available potassium (**E**) for the 2011–2015 time period. The relationship between available phosphorus and available potassium (**F**) for the 2016–2020 time period. The relationship between available nitrogen and available potassium (**G**) for the 2006–2010 time period. The relationship between available nitrogen and available potassium (**H**) for the 2011–2015 time period. The relationship between available nitrogen and available potassium (**I**) for the 2016–2020 time period. The gray open circles represent the element concentration, and the broken lines represent the relationship between the elements.

## 4. Discussion

The significance of this research is to provide support for better evaluation of soil fertility, the guidance of nutrient management, and understanding of the biological earth cycle through the study of the temporal and spatial variability of available N, P, and K stoichiometry in farmland soil. It is of great significance to study the temporal and spatial variation characteristics of available nitrogen, phosphorus, and potassium content in soil for exploring nutrient cycles and ecological environment change in soil circle, rational utilization of soil resources, and improvement of the ecological environment [40].

Understanding the distribution of soil nutrients and their influencing factors is of great significance to fertilizer management and environmental protection [41].

The average values of available N (84.22 mg/kg = 6.02 mmol/kg) and P (30.15 mg/kg = 0.97 mmol/kg) concentrations in farmland soils in Hainan Island during the period 2006–2020 were lower than the average N (94.4 mmol/kg) and P (14.9 mmol/kg) concentrations in global farmland soils [22]. This was due to the influence of environmental factors in terrestrial ecosystems such as climate, parent material, topography, soil properties, vegetation types, human activities, etc., resulting in large spatial heterogeneity of soil nutrients and their stoichiometric characteristics at regional and global scales [42]. According to the Chinese soil nutrient classification standard, available nitrogen and available potassium in the study area are at medium level, and available phosphorus was at high levels. Shangguan mentioned in his study entitled "A China Data Set of Soil Properties for Land Surface Modeling" that the available nitrogen, phosphorus, and potassium contents of surface soil in China ranged from 9.8–648 mg/kg, 0.9–16.5 mg/kg, and 21.9–703.8 mg/kg, respectively [43]. It can be seen that the content of available nitrogen and available potassium in the farmland soil of Hainan Island were both within this range, but the content of available phosphorus was higher than that of the topsoil in China.

Our research results showed that the available N, P, and K contents in farmland soil of Hainan Island were the highest during 2016–2020 and the lowest during 2010–2015. Since the 1980s, the soil nitrogen and phosphorus content have increased significantly [29]. Our results also showed that the concentration of nitrogen and phosphorus in the 2016–2020 time period was significantly higher than the other two time periods. Paradoxically, the concentration of N, P, and K during the period 2006–2010 was significantly higher than the concentration of N, P, and K during the period 2011–2015. This may be related to the amount of fertilizer applied at different stages. The changes in nitrogen, phosphorus, and potassium content in farmland soil were closely related to fertilization [44]. On the other hand, the concentrations of nitrogen, phosphorus, and potassium in soil showed different rules in different time periods, which may be related to continuous cropping in farmland soil. Some scholars have shown that continuous cropping will cause an imbalance of soil elements [45,46].

Studies have shown that the N:P ratio of surface soil has increased significantly from the 1980 s (2.9) to 2000 s (3.5) [47,48]. However, in our research, the N/P of farmland soil in Hainan Island showed no significant difference in the three time periods of 2006–2010, 2011–2015, and 2016–2020. This was diametrically opposed to the results of studies before 2000. It was not true for N/K, which showed a gradually increasing trend. The ratio of P and K also reached its maximum value in 2016–2020. Phosphorus and potassium are mainly controlled by the weathering rate of the parent material [49,50]. Therefore, the difference in the ratio between different elements at different stages may be related to the weathering rate of parent materials at different stages.

Studying the temporal and spatial variability of soil nutrients in tropical farmland was of great significance for promoting the sustainable development of tropical agriculture [29]. The spatial heterogeneity of soil was a kind of soil characteristic widely existing in soil aggregates [51], soil nutrients [52], and soil moisture [53]. Many studies have shown that soil organic carbon, soil moisture, and other indicators were different in spatial distribution [54,55]. This study mainly discussed the spatial variability of soil nutrients. Due to various natural and man-made factors, soil nitrogen, phosphorus, and potassium were unevenly distributed in space [41]. The available nitrogen, phosphorus, and potassium contents of Ding'an, Haikou, Wuzhishan, Changjiang, Dongfang, Ledong, Sanya, and Lingshui in the study area are all within the range of previous research results [56–59]. No matter what time period in Wenchang and Haikou, farmland soil P and K content were higher. This was similar to the results of the second soil survey on farmland soil in Hainan Island. The results showed that soil available N, P, and K contents changed in different degrees in different time periods. The spatial variability of soil elements is determined by both internal and external factors. The temporal and spatial differences

of soil element content were not only related to the different soil parent materials, soil types, vegetation cover, and land-use patterns in this region, but also closely related to the content of initial soil elements, generally speaking, the higher the content of initial soil elements, the faster the loss of nutrient elements [54]. Studies have shown that soil parent material is the main factor affecting the spatial distribution pattern of soil phosphorus and potassium [41]. The spatial distribution of available nutrients is also dependent on temporal variability. With the increase of years, the available nitrogen content in Haikou showed an increasing trend, the available phosphorus content showed an increasing trend in the south of Wenchang, and the available potassium in Tunchang and the southeast of Haikou showed an increasing trend. The temporal and spatial variability of soil nutrients could also be caused by topographic factors, which have a strong relation with soil nutrients [60], thereby affecting nutrient stoichiometry. At the same time, it may also be closely related to the different types of vegetation in farmland soils [61] in different regions of Hainan Island. Wu et al. (2016) believed that vegetation type was the dominant controlling factor of soil nitrogen. Furthermore, available nitrogen, phosphorus, and potassium were the most easily absorbed nutrients by plants, so they were greatly affected by plant growth [41]. Therefore, the different types of vegetation in different regions will lead to different absorption of available N, P, and K by plants, which in turn will affect the content of N, P, and K in the regional soil. One of the most important factors affecting the content of elements in the soil of different regions was the different ways of using land by humans in different regions [18], and the application of fertilizers was a very important aspect. Studies have shown that human land use will significantly increase the potassium content of the soil [18]. The changes in soil nutrient content in recent years were closely related to the application of chemical fertilizers, and the uneven distribution of soil nutrient content was a reflection of the application of fertilizers in the study area [29]. Therefore, fertilization planning can be carried out according to the spatial and temporal distribution of soil nutrients to maintain the balance of soil nutrient content.

Many studies have shown that there was a relation between soil N, P, and K and many soil properties. For example, studies have pointed out that there was a positive relationship between soil organic matter and soil nitrogen, phosphorus, and potassium nutrients [62] and soil available N, P, and K had significant positive effects on the contents of N, P, and K in potato tubers [63]. Moreover, Mousumi et al. showed that there was a close relationship between soil nitrogen, phosphorus, and potassium and soil organic carbon [64]. This study found that there was also a relation between soil nitrogen, phosphorus, and potassium itself. Moreover, the relationship between soil available phosphorus and potassium in different time periods was not the same. However, available N and P, and available N and K have the same relationship in different time stages. In their research, Oimahmad et al. showed that there was a strong relationship between elements such as nitrogen, phosphorus, and potassium in the soil of the mining area [65]. This study further showed that there was also a relation between the available nitrogen, phosphorus, and potassium content of farmland soil, which was the same as the findings in the soil in mining areas.

## 5. Conclusions

The content of available nitrogen, phosphorus, and potassium in farmland soils in Hainan Island showed different patterns of temporal and spatial variation. The content of available N, P, and K showed the same pattern of variation over time, which was the highest in the 2016–2020 time period and the lowest in the 2010–2015 time period. There is no significant difference in N/P in different time stages, while the maximum values of N/K and P/K both appeared in the 2016–2020 time period. In the 2006–2010, 2011–2015, and 2016–2020 stages, the available nitrogen content first decreased and then increased with the increase of the available phosphorus content, while the available phosphorus content in the 2011–2015 and 2016–2020 periods decreased firstly and then increased with the increase of the available potassium content. The available nitrogen content in the three time periods first increased and then decreased with the increase of available potassium content. There

are similar trends in the spatial distribution of available nitrogen, available phosphorus, and potassium at different times. The content of N increased in Haikou, Lingao, and Ding'an, and P increased in Wengchang and Lingshui and K increased in Danzhou, and Wanning as time increased. Combining tropical crop types and soil nutrient status, our research will focus on crop nutrient management and fertilization regulation in the future.

**Author Contributions:** G.F. designed the research. C.J. conceived and designed the research and wrote the manuscript. H.L. performed the main experimental work. Z.C. and Q.L. performed the data analysis. All authors have read and agreed to the published version of the manuscript.

**Funding:** This research was funded by the earmarked fund for China Agriculture Research System (CARS-33-ZP2) and the National Key R&D Program of China (2020YFD1000600).

**Institutional Review Board Statement:** Not applicable.

**Informed Consent Statement:** Not applicable.

**Data Availability Statement:** Data is contained within the article.

**Acknowledgments:** The authors are grateful to China Rubber Research Institute for providing the data.

**Conflicts of Interest:** The authors declare that they have no conflict of interest.

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
