# Peer review of "Spatial-Temporal Variation of N, P, and K Stoichiometry in Cropland of Hainan Island"

_agriculture, doi:10.3390/agriculture12010039_

Round 1

Reviewer 1 Report

Paper "Spatial-temporal variation of N, P, K stoichiometry in cropland of Hainan Island" is very interesting.

Authors analyzed differences in the element content of N, P, and K under different time and space conditions as well as the relationships between N, P, K.

Analyses was made for five separate farmland soil samples.

Authors used the average values of elements in the three time periods from 2006 to 2010, 2011 to 2015, 2016 to 2020. Why? This is incorrect. Statistical analyses should be performed for all years! Needs correction.

"The density histogram was used to study the influence of different time periods on the content of farmland soil N, P and K." Not. "Density histogram" is for presentation of distribution of data. For "influence" Authors should be use regression analysis.

In Figures 2 and we observed lack of nornal distributions for all observed traits. in these cases analysis of variance without transformation data is inncorrect.

Subsection 3.3. and Figure 5: "the broken lines represent the correlation between the elements". Not!!! These lines represent regression, not correlation. Authors should be estimates correaltion coefficients and testing these values.

Author Response

Point 1: Authors used the average values of elements in the three time periods from 2006 to 2010, 2011 to 2015, 2016 to 2020. Why? This is incorrect. Statistical analyses should be performed for all years! Needs correction.

Response 1:Accepted. Thank you for your suggestion. The reasons why we choose five years as a time period: first, the five-year data sampling point basically covers an area, and second, it is related to the five-year agricultural plan and policies of the Chinese government.In order to better understand the changes of data, we analyzed and added an annual change figure.

Figure 1 The content of available nitrogen (A) at all years. The content of available phosphorus (B) at all years. The content of available potassium (C) at all years. The stoichiometric ratio of N, P(D) at all years. The stoichiometric ratio of N, K (E)at all years. The stoichiometric ratio of P, K(F) at all years.The content of available nitrogen, phosphorus and potassium all showed an increasing trend after 2015. No obvious trend is observed between N/P and N/K in each year, and further analysis and verification are needed. P/K also showed an increasing trend since 2015.Please find our detailed revision to the figure1 in the revision.

Point 2: "The density histogram was used to study the influence of different time periods on the content of farmland soil N, P and K." Not. "Density histogram" is for presentation of distribution of data. For "influence" Authors should be use regression analysis.

Response 2: Accepted.The density histogram was used to study the frequency and the range of  data. Please find our detailed revision to the 2.3 in the revision.

Point 3: In Figures 2 and we observed lack of nornal distributions for all observed traits. in these cases analysis of variance without transformation data is inncorrect.

Response3:Accepted. Thank you for your suggestion. You are right. The data should be transformed before statistical analysis. To improve normality of distributions, the data of the content N, P and K and the ratio of the N, P and K have been transformed by natural logarithm before all statistical analysis. The Density histograms of N, P and K at different stages and the density histogram of the ratio of N, P, K at different stages have been re-presented in our manuscript. Please find our detailed revision (Figure 4 and Figure 5).

Point 4: Subsection 3.3. and Figure 5: "the broken lines represent the correlation between the elements". Not!!! These lines represent regression, not correlation. Authors should be estimates correaltion coefficients and testing these values.

Response4:  Accepted. Thank you for your suggestion. You are right.The broken lines represent the correlation between the elements. Please find our detailed revision to the 3.3 and figure6 in the manuscript.

Reviewer 2 Report

The current study entitled “Spatial-temporal variation of N, P, K stoichiometry in cropland of Hainan Island” is good. For a better understanding in-depth, it is a need for time to work on this topic. Furthermore, the achievement of potential benefits by using current technology is also dependent on the extensive research work for more exploration. Although the experiment is well organized, yet I suggest a rejection due to the following deficiencies.

Major Concerns

  • Systematic abstract is missing. Introduce the need for study in 1-2 lines. i.e., Spatial-temporal variation of N, P, K
  • Please give a clear cut point problem source as a problem statement that is tackled in the current study.
  • Give logical reason for the selection of current strategy i.e., Spatial-temporal variation of N, P, K stoichiometry.
  • Quantitative data is also important to support your conclusion. Would you please provide some quantitative data in terms of percentage significant increase or decrease in the abstract?
  • Please provide a conclusive conclusion with is withdrawn through research in a single line. The statement “There are strong variations on the spatial distribution of available nitrogen, available phosphorus and potassium in different times” is general. Please conclude with a statement that shows a knowledge gap covered, potential beneficiaries and specific recommendations as well.
  • Give future prospective in a single line.
  • As per standard suggestions, please avoid using title words as keywords
  • Please follow the title in the introduction section, i.e., the spatial distribution of available nitrogen, available phosphorus and potassium, then a temporal variation of N, P, K stoichiometry, the importance of current study in cropland of Hainan Island knowledge gap, hypothesis and aims.
  • Also, provide a novelty statement at the end. What new things authors have done or correlated in this research compared to old ones?
  • Would you please give a single line about the knowledge gap which your research has covered along with the hypothesis statement?
  • Material and methods are ok.
  • In the result, each figure must be self-explanatory. Please provide the details of abbreviations A,B,C…. at the end of each figure.
  • Please give a conclusive conclusion.
  • If the authors are not sure, then give future recommendations for more research and investigation.
  • Add the targeted beneficiary audience who will get benefits from this research.
  • Also, give clear-cut recommendations and future prospective regarding this research.

Author Response

Point 1: Systematic abstract is missing. Introduce the need for study in 1-2 lines. i.e., Spatial-temporal variation of N, P, K

Response 1: Accepted. Thank you for your suggestion. Studying the temporal and spatial variability of soil nutrients in tropical farmland was of great significance for exploring the variation of soil nutrients and promoting the sustainable development of tropical agriculture. And we had added it to the abstract  in the manuscript.

Point 2: Please give a clear cut point problem source as a problem statement that is tackled in the current study.

Response 2: Accepted. Thank you for your suggestion.The ecological stoichiometry theory about tropic soils on spatial-temporal variation is lacked. Based on this study, we can explore the spatial and temporal variation of soil nutrients in Hainan Island, find out the soil nutrient restriction factors and explore the soil nutrients in Hainan Island potential.

Point 3: Give logical reason for the selection of current strategy i.e., Spatial-temporal variation of N, P, K stoichiometry.

Response3: Accepted. Thank you for your suggestion. We  choose the topic “Spatial-temporal variation of N, P, K stoichiometry in cropland of Hainan Island” for an understanding of Spatial-temporal variation of N, P, K stoichiometry in cropland of Hainan Island can help land managers choose effective methods to improve cultivation management which limit the crop yield and quality.

Point 4: Quantitative data is also important to support your conclusion. Would you please provide some quantitative data in terms of percentage significant increase or decrease in the abstract?

Response4:  Accepted. Thank you for your suggestion.The content of N increased in Haikou、Lingao、Dingan and P increased in Wengchang、Lingshui and K increased in Danzhou、Wanning as time increasing.  Please find our detailed revision to the abstract  in the manuscript.

Point 5: Please provide a conclusive conclusion with is withdrawn through research in a single line. The statement “There are strong variations on the spatial distribution of available nitrogen, available phosphorus and potassium in different times” is general. Please conclude with a statement that shows a knowledge gap covered, potential beneficiaries and specific recommendations as well.

Response5:  Accepted. Thank you for your suggestion. There are similar trendcy on the spatial distribution of available nitrogen, available phosphorus and potassium in different times. The content of N increased in Haikou、Lingao、Dingan and P increased in Wengchang、Lingshui and K increased in Danzhou、Wanning as time increasing.  And we had added it to the conclusion  in the manuscript. Please find our detailed revision  in the manuscript.

Point 6: Give future prospective in a single line.

Response6: Accepted. Thank you for your suggestion. Combining tropical crop types and soil nutrient status, our research will focus on crop nutrient management and fertilization regulation in the future. And we had added it to the end  in the manuscript. Please find our detailed revision  in the manuscript.

Point 7: As per standard suggestions, please avoid using title words as keywords

Response 7:  Accepted. Thank you for your suggestion. Hainan Island; soli nutrient ; Spatial-temporal variability; stoichiometry. And we had corrected it to the keywords in the manuscript. Please find our detailed revision  in the manuscript.

Point 8: Please follow the title in the introduction section, i.e., the spatial distribution of available nitrogen, available phosphorus and potassium, then a temporal variation of N, P, K stoichiometry, the importance of current study in cropland of Hainan Island knowledge gap, hypothesis and aims.

Response 8:  Accepted. Thank you for your suggestion. Studying the temporal and spatial variability of soil nutrients in tropical farmland was of great significance for exploring the variation of soil nutrients and promoting the sustainable development of tropical agriculture.we had added to the introduction in the manuscript. Please find our detailed revision  in the manuscript.

Point 9: In the result, each figure must be self-explanatory. Please provide the details of abbreviations A,B,C…. at the end of each figure.

Response 9:  Accepted. Thank you for your suggestion. we had added the A,B,C….explanations at the end of each figure(figure2-4.figure6) . Please find our detailed revision  in the manuscript.

Reviewer 3 Report

The article related to spatial-temporal variation and stoichiometry follows the expected structure of this type of articles. I have some doubts about the text and I believe that some responses are necessary from authors.

The first one is the consideration of Ecological stoichiometry as a science, I disagree as it is part of Ecology or Agricultural Chemistry or Plant Nutrition, among others. If this is from Sterner (2002), I understand the citation of the authors in the introduction, but I disagree with this affirmation, maximum if the authors include a Justus von Liebig, the father of the Agricultural Chemistry. As a suggestion, authors can check the references of Epstein, Marschner, Corley, Mengel and Kirkby, Barceló and others about mineral nutrition and essential elements.

The novelty, considering the publications done in previous years (from the middle of the past century to its end) is the incorporation of the spatial variability, but nit the temporal variation in soils and plants.

Regarding to soil description given in materials and methods, as a suggestion, the use of one of the major systems to classify soils would be necessary, Soil Taxonomy or WRB. Moreover, soil samples were “The air-dried soil samples were passed through 1mm and 0.25mm sieves”. This size to sieve the soil samples is not a standard methodology to determine soil nutrients availability as it is commonly used 2 mm sieve.

Poor information about the date when soil samples were taken and how authors obtained data for a five years period. How many soil samples were taken?

The methods to determine the availability of N, P and K are not well described and referenced.

The data has not been tested regarding the normality of them, or it was done but no reference about the method is given.

Results and Discussion is a sequence of the description of the situation, indicating that this article is so close to a case study.

Finally, the conclusions are of regional interest and as a suggestion, it could be specified on results with future projections and their use in other places.

Please check the following:

As a suggestion, to facilitate the reading of the text, check the separation of the words and correct it if possible, i.e. line 9: “microorgan-isms maybe microorga-nisms”, at the end of the lines 9, 10, 35, 70

Check the format of the citation in the text, the size of the numbers into brackets. I think that the size would be higher following the template of the journal.

Check some sentences about English grammar, ie.e “Locate each sampling site with GPS and record” (line 113) and other lines like line 125….

Author Response

Point 1: The first one is the consideration of Ecological stoichiometry as a science, I disagree as it is part of Ecology or Agricultural Chemistry or Plant Nutrition, among others. If this is from Sterner (2002), I understand the citation of the authors in the introduction, but I disagree with this affirmation, maximum if the authors include a Justus von Liebig, the father of the Agricultural Chemistry. As a suggestion, authors can check the references of Epstein, Marschner, Corley, Mengel and Kirkby, Barceló and others about mineral nutrition and essential elements.

Response 1: Accepted. Thanks the reviewer for your efforts and comments in our manuscript. We understand the concerns of the reviewers and carefully reviewed the articles on mineral nutrition and essential elements provided by the reviewers. What makes us feel excited is how much we benefit from these articles! Moreover, based on the new perspectives on thinking about the problem provided to us by the reviewers, we have revised and embellished the descriptions of stoichiometry and mineral elements in our manuscript.

It is the following:

The stoichiometry provides a frame-work approach to understanding the function of ecosystems and the balance of elements [1]. The development and suitability of plants largely depends on the availability of mineral elements in the soil [2]. Nitrogen (N), phosphorus (P) and potassium (K) as important mineral elements are one of the core focuses of ecosystem ecology [3].

Certainly, we would like to revise it further for improvement the quality of our manuscript if the reviewer considers it as necessary.

Point 2: Regarding to soil description given in materials and methods, as a suggestion, the use of one of the major systems to classify soils would be necessary, Soil Taxonomy or WRB. Moreover, soil samples were “The air-dried soil samples were passed through 1mm and 0.25mm sieves”. This size to sieve the soil samples is not a standard methodology to determine soil nutrients availability as it is commonly used 2 mm sieve.

Response 2: Accepted. Thank you for your suggestion.The main soil types of Hainan Island include ferrisols,allitic soils,alfisols, cambosols, andosols, anthrosols, halosols and entisols.The air-dried soil samples were passed through 2mm sieves and then the sieved soil samples were mixed uniformly and put into a zip lock bags.We  had corrected in the article in the 2.2. Please find our detailed revision  in the manuscript.

Point 3: Poor information about the date when soil samples were taken and how authors obtained data for a five years period. How many soil samples were taken?

Response3: Accepted. Thank you for your suggestion.From 2006 to 2020, soil samples were collected annually, and a total of 5,877 soil samples were collected. The distribution of sampling sites took into account the land use throughout Hainan Island. Please find our detailed revision  in the manuscript.

Point 4: The methods to determine the availability of N, P and K are not well described and referenced.

Response4:  Accepted. Thank you for your suggestion.The soil available nitrogen was determined by alkaline hydrolysis distillation method.Soil available P was measured by colorimetric method after hydrochloric acid-ammonium fluoride extraction.Available K was determined using flame photometry after ammonium acetate extraction.And we had corrected to the 2.2 in the article. Please find our detailed revision  in the manuscript.

Point 5: The data has not been tested regarding the normality of them, or it was done but no reference about the method is given.

Response5:  Accepted. Thank you for your suggestion.To improve normality of distributions, the data of the content N, P and K and the ratio of the N, P and K were log 10 -transformed(Figure 3, Figure 4) for permutation test, but shown in untransformed values for easier understanding in our article.The  Density histograms of N, P and K at different stages(Figure 3) and  the density histogram of the ratio of N, P, K at different stages (Figure 4) are corrected in the manuscript. Please find our detailed revision  in the manuscript.

Point 6: Finally, the conclusions are of regional interest and as a suggestion, it could be specified on results with future projections and their use in other places.

Response 6:  Accepted. Thank you for your suggestion.The content of N increased in Haikou、Lingao、Dingan and P increased in Wengchang、Lingshui and K increased in Danzhou、Wanning as time increasing. Combining tropical crop types and soil nutrient status, our research will focus on crop nutrient management and fertilization regulation in the future. And we had added it to the conclusion in the manuscript. Please find our detailed revision  in the manuscript.

Point 7: As a suggestion, to facilitate the reading of the text, check the separation of the words and correct it if possible, i.e. line 9: “microorgan-isms maybe microorga-nisms”, at the end of the lines 9, 10, 35, 70

Response 7:  Accepted. Thank you for your reminding. We have carefully examined the full manuscript once again.We found some words separated because of  system typesetting and automatically generated.

Point 8: Check the format of the citation in the text, the size of the numbers into brackets. I think that the size would be higher following the template of the journal.

Response8:  Accepted. Thank you for your reminding. We have carefully examined the full manuscript once again.We found the format and the number’s size problems   because of  system typesetting and automatically generated.

Point 9: Check some sentences about English grammar, ie.e “Locate each sampling site with GPS and record” (line 113) and other lines like line 125….

Response 9:  Accepted. Thank you for your suggestion.The geographical location of the sample points were Located by GPS.  The longitude and latitude was recorded in detail. we had corrected the line113 and 125 in the manuscript. Please find our detailed revision  in the manuscript.

Round 2

Reviewer 1 Report

Thank you for your corrections. Now, is ok.

Author Response

Dear Reviewer:

Thanks again for your careful inspection and good suggestions.

Reviewer 2 Report

Dear Authors

I am satisfied with the changes made.

Author Response

(The authors gave the same response as above.)

Reviewer 3 Report

It is important to thanks author for improving the article. However, it needs more corrections and clarifications.

Introduction has low or nothing about the agricultural chemistry as a source of the study of plant nutrition and nutrient availability in soils. As a suggestion, this would be mentioned as some references were given in previous report.

As a suggestion, the importance and projection of the work to other situations would help to increase the interest of readers.

Point 2.1. Which classification system was used to describe the soil types? Need reference.

Point 2.2. Please, add the references of the analytical methods used.

Authors have changed the size of the fine earth used from 1 mm to the previous version to 2mm in the last one. May I have to understood that it was a previous mistake? I hope so.

Check the following minor mistakes and correct to improve the text (I think it is necessary):

Lines 10-11: “el-ements”

Lines 23-24: “The content of N 23 increased in Haikou、Lingaod in en and P increased in Wengchang、Lingshui and K increased in 24 Danzhou、Wanning as time increasing.”

Lines 93-94: “var-iation”

Line 104: “sols,allitic soils,alfisols”

Line 118: “lat-itude”

Line 140: “ln-transformed(Figure”

Line 198: check the free space given between paragraphs

Line 280: “Figure5.” And the quality of the figure, is a bit low.

Lines 475-477: “The content of N increased in Hai-475 kou、Lingao、Dingan and P increased in Wengchang、Lingshui and K increased in 476 Danzhou、Wanning as time increasing.”              

Author Response

Point 1: Introduction has low or nothing about the agricultural chemistry as a source of the study of plant nutrition and nutrient availability in soils. As a suggestion, this would be mentioned as some references were given in previous report.

Response 1: Accepted. Thank you for  your careful inspection.Although some studies have gradually applied ecological stoichiometry to crop soils in recent years. These studies have mainly focused on the effects of different fertilization methods[22,23], different environmental stresses [24], and different growth stages[25] on the ecological stoichiometry of crops.There is a lack of research on farmland soil stoichiometry that includes both temporal and spatial variation. Please find our detailed revision to the introduction in the manuscript.

Point 2: Which classification system was used to describe the soil types? Need reference.

Response 2: Accepted. Thank you for  your suggestion.Our reference is that “Soil System Classification Research Group of Nanjing Institute of Soil Research, Chinese Academy of Sciences, etc. Classification of Soil System in China (Amendment Scheme)[M].Beijing: China Agricultural Science and Technology Press,1995:5-23.” Please find our detailed revision in the manuscript.

Point 3: Please, add the references of the analytical methods used.

Authors have changed the size of the fine earth used from 1 mm to the previous version to 2mm in the last one. May I have to understood that it was a previous mistake? I hope so.

Check the following minor mistakes and correct to improve the text (I think it is necessary):

Response 3: Accepted. Thank you for  your careful inspection.Please find our the references of the analytical methods used in our manuscript.

Shen R P,Sun B,Zhao Q G. Spatial and temporal variability of N,P

and K balances for agro-ecosystems in China[J]. Pedosphere,2005,15(3): 347 -355.

Lu Rukun, Shi Zhengyuan, Shi Jianping. Evaluation of farmland nutrient balance in 6 provinces in southern China Price and dynamic change studies [J]. Agricultural Science of China, 2000,33 (2): 63-67.

We are sorry that we had made a mistake and We confounded the treatment of soil sieve 1mm before organic matter determing at the beginning.

Point 4: Lines 10-11: “el-ements”

Lines 23-24: “The content of N 23 increased in Haikou、Lingaod in en and P increased in Wengchang、Lingshui and K increased in 24 Danzhou、Wanning as time increasing.”

Lines 93-94: “var-iation”

Line 104: “sols,allitic soils,alfisols”

Line 118: “lat-itude”

Line 140: “ln-transformed(Figure”

Line 198: check the free space given between paragraphs

Line 280: “Figure5.” And the quality of the figure, is a bit low.

Lines 475-477: “The content of N increased in Hai-475 kou、Lingao、Dingan and P increased in Wengchang、Lingshui and K increased in 476 Danzhou、Wanning as time increasing.” 

Lines 10-11: “el-ements”

Lines 23-24: “The content of N 23 increased in Haikou、Lingaod in en and P increased in Wengchang、Lingshui and K increased in 24 Danzhou、Wanning as time increasing.”

Lines 93-94: “var-iation”

Line 104: “sols,allitic soils,alfisols”

Line 118: “lat-itude”

Line 140: “ln-transformed(Figure”

Line 198: check the free space given between paragraphs

Line 280: “Figure5.” And the quality of the figure, is a bit low.

Lines 475-477: “The content of N increased in Hai-475 kou、Lingao、Dingan and P increased in Wengchang、Lingshui and K increased in 476 Danzhou、Wanning as time increasing.” 

Response 4: Accepted. Thank you for  your careful and serious inspection.We adopt  your suggestion and examined the article carefully once again. We modified the spelling details in the manuscript item by item. Please find our detailed revision  in the manuscript.

Lines 10-11: “elements”

Lines 23-24: “The content of N 23 increased in Haikou、Lingaod 、Ding’an and P increased in Wengchang、Lingshui and K increased in Danzhou、Wanning as time increasing.”

Lines 93-94: “variation”

Line 104: The main soil types of Hainan Island include ferrisols, ferralsols,alfisols, cambosols, andosols, anthrosols, halosols and entisols.

Line 118: “latitude”

Line 140: “log10-transformed

Line 198: we checked and corrected the free space given between paragraphs

Line 280: The spatial distribution of each indicatoravailable nitrogen (A) at different stages.The spatial distribution of each indicator available phosphorus(B) at different stages.The spatial distribution of each indicatoravailable potassium (C)  at different stages.

Lines 475-477: “The content of N increased in Haikou、Lingao、Ding’an and P increased in Wengchang、Lingshui and K increased in Danzhou、Wanning as time increasing.” 

Dear Reviewer:Thanks again for your careful inspection and good suggestions. Even if we had revised the format of the words , the mistakes are existing because of  system typesetting and automatically generated.We contact with the  Agriculture Editorial Office to resolve these problems.
